# Multi-Turn Code Generation Through Single-Step Rewards

**Arnav Kumar Jain** [* 1 2]   **Gonzalo Gonzalez-Pumariega** [* 3]   **Wayne Chen** [3]   **Alexander M Rush** [3]
**Wenting Zhao** [† 3]   **Sanjiban Choudhury** [† 3]

## Abstract

We address the problem of code generation from multi-turn execution feedback. Existing methods either generate code without feedback or use complex, hierarchical reinforcement learning to optimize multi-turn rewards. We propose a simple yet scalable approach, $\mu$CODE, that solves multi-turn code generation using only single-step rewards. Our key insight is that code generation is a one-step recoverable MDP, where the correct code can be recovered from any intermediate code state in a single turn. $\mu$CODE iteratively trains both a generator to provide code solutions conditioned on multi-turn execution feedback and a verifier to score the newly generated code. Experimental evaluations show that our approach achieves significant improvements over the state-of-the-art baselines. We provide analysis of the design choices of the reward models and policy, and show the efficacy of $\mu$CODE at utilizing the execution feedback. Our code is available here.

## 1. Introduction

Software engineers often iteratively refine their code based on execution errors. A common strategy for machine code generation is thus to repair code using execution feedback at test time (Chen et al., 2024; Wang et al., 2024b; Zhao et al., 2024). However, prompting alone is insufficient as it cannot teach how to recover from all possible errors within a limited context.

We need to train models that can learn from execution feedback during training. Existing approaches fall into either single-turn or multi-turn settings. In the single-turn setting, methods either train without execution feedback (Zelikman et al., 2022) or perform one-step corrections (Welleck et al.,

2023; Ni et al., 2024). However, these struggle to iteratively correct errors over multiple turns. Multi-turn approaches, on the other hand, rely on complex reinforcement learning (RL) (Gehring et al., 2024a; Kumar et al., 2024b; Zhou et al., 2024) to optimize long-term rewards. While effective in principle, these methods suffer from sparse learning signals which makes learning inefficient.

Our key insight is that code generation is *a one-step recoverable Markov Decision Process (MDP), implying that the correct code can be recovered from any intermediate state in a single step*. This allows us to greedily maximize a one-step reward instead of relying on complex multi-step reward optimization. As a result, this reduces the problem from reinforcement learning, which requires exploration and credit assignment, to imitation learning, where the model simply learns to mimic correct code, leading to a more stable and efficient training process.

We propose $\mu$CODE, a simple and scalable approach for multi-turn code generation from execution feedback. During training, $\mu$CODE follows an *expert iteration* (Anthony et al., 2017) framework with a *local search expert*, enabling iterative improvement of both the generator and the expert. The process begins by rolling out the current code generator to collect interaction data with execution feedback. A single-step verifier is then trained on this data and utilized to guide a local search expert in refining the code and generating training labels. Finally, the generator is fine-tuned using these labels. Given recent trends of test-time scaling in generating high quality solutions (Brown et al., 2024; Snell et al., 2024; Wu et al., 2024), $\mu$CODE also uses the learned verifier for inference-time scaling. Here, $\mu$CODE samples $N$ trajectories; at each step, $\mu$CODE picks the best code solution ranked by the learned verifier.

The key contributions of this work are as follows:

1. A novel framework, $\mu$CODE, for training code generators and verifiers through multi-turn execution feedback. We add theoretical analysis of performance bounds using the property of one-step recoverability for this task.
2. We propose a *multi-turn Best-of-N (BoN) approach* for inference-time scaling and present benefits of learned verifier to select the code solution at each turn.
3. Our approach $\mu$CODE outperforms leading multi-turn

---

[*]Equal contribution  [1]Mila- Quebec AI Institute  [2]Université de Montréal  [3]Cornell University. Correspondence to: Arnav <arnavkumar.jain@mila.quebec>, Gonzalo <gg387@cornell.edu>.

*Proceedings of the $42^{nd}$ International Conference on Machine Learning*, Vancouver, Canada. PMLR 267, 2025. Copyright 2025 by the author(s).

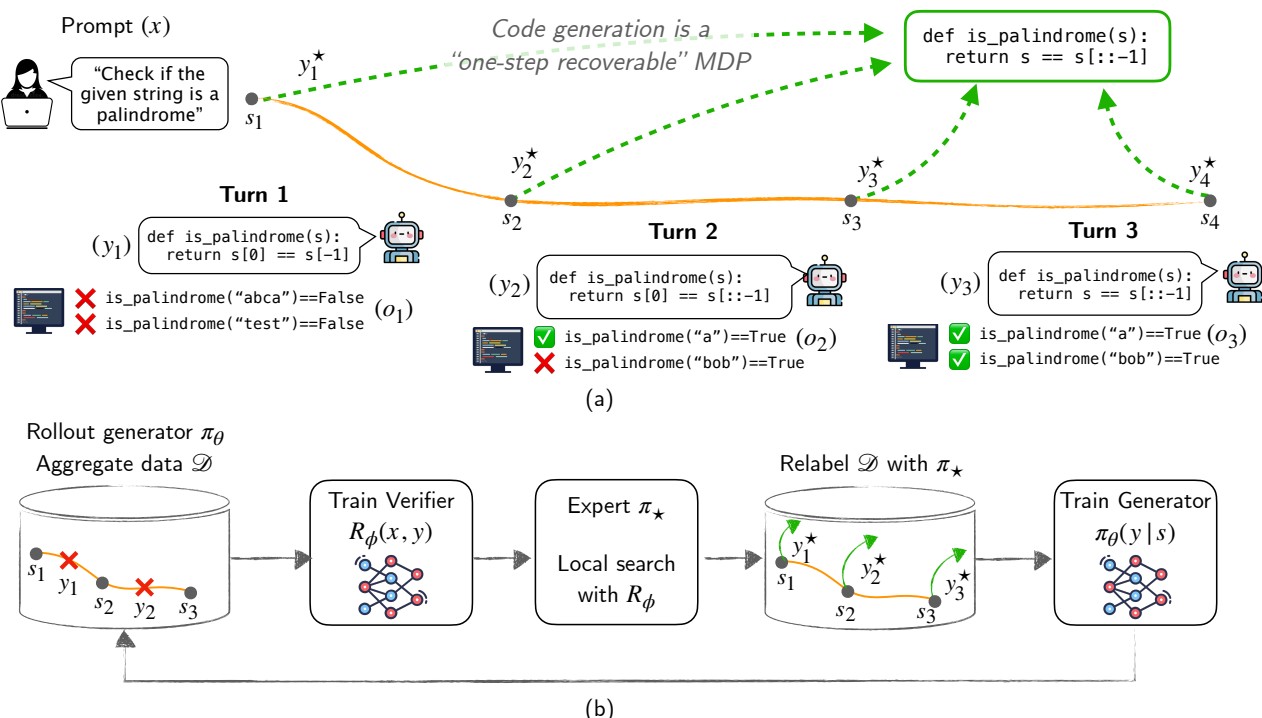

*Figure 1.* (a) We define the task of multi-turn code generation where for an initial problem $x$, the generator $\pi_\theta$ provides a solution $y_1$. This solution is evaluated with the public test to get execution feedback $o_1$. At a turn $t$, the generator is conditioned on the history to generate solution $y_t \sim \pi_\theta(.|x, y_{<t}, o_{<t})$. The rollout ends when the turn limit is reached or the public tests pass upon which the solution is executed on private tests. Since, the agents can generate the optimal solution at any turn, this is a 1-step recoverable process. (b) Training loop of our method $\mu$CODE – which comprises of a generator and a learned verifier. During each iteration, rollouts are collected using $\pi_\theta$ and we train a verifier $R_\phi$ to rank candidate solutions for a prompt. The verifier $R_\phi$ is then used to construct a local expert and relabel the collected rollouts. Lastly, the generator is fine-tuned with this expert dataset.

approaches on MBPP (Austin et al., 2021), HumanEval (Chen et al., 2021) and CodeContests (Li et al., 2022a) benchmarks. Our ablations show that learned verifiers aid in learning better generators and show promising scaling law trends with higher inference budgets.

## 2. Background

In multi-turn code generation, an agent iteratively refines a program to maximize its correctness on private test cases. Given an initial problem prompt $x$, at each turn $t$, the agent generates a complete code snippet $y_t$ and executes it on a set of public tests. The outcomes $o_t$ from these tests serve as observations that guide subsequent refinements. This process continues until the agent generates a code snippet $y_t$ that passes all public tests, at which point the episode terminates, or until the maximum number of turns $T$ is reached without success. The first successful code, $y_t$, is then evaluated on private tests to compute the correctness score $C(x, y_t) \in \{0, 1\}$.

We model this as a Markov Decision Process (MDP),

where the state is the interaction history $s_t = \{x, y_1, o_1, \ldots, y_{t-1}, o_{t-1}\}$ where $s_1 = \{x\}$, and the action is the code snippet $y_t$. The oracle reward is defined as $R(s_t, y_t) = R(x, y_t) = C(x, y_t)$ if $y_t$ passes all public and private tests (terminating the episode), or 0 otherwise.

During training, given a dataset of problem prompts $\mathcal{D}$, the goal is to find a generator $\pi_\theta(y_t|x, y_1, o_1, \ldots, y_{t-1}, o_{t-1})$, that maximizes the cumulative discounted reward $R(x, y_t)$:

$$\max_{\pi_\theta} \mathbb{E}_{x \sim \mathcal{D}, y_t \sim \pi_\theta(\cdot|s_t)} \left[ \sum_{t=1}^{T} \gamma^t R(x, y_t) \right], \quad (1)$$

where $\gamma \in [0, 1)$ is the discount factor. As shown in Eq. 1, the objective optimizes for a policy to generate the correct solution with as few turns as possible. However, at any step $t$, the agent can generate the correct code solution $y_t = y^\star$ such that $C(x, y^\star) = 1$ (as shown in Fig. 1 (a)) – a *one-step recoverable* process. In the following section, we describe $\mu$CODE, a simple and scalable framework that leverages the one-step recoverability and reduces the problem of reinforcement learning to imitation learning.

**Algorithm 1** $\mu$CODE: Training

---

**input** Initial generator $\pi_0$, multi-turn code environment $\mathcal{E}$, and max iterations M

1: **for** iteration i = 1 … M **do**
2:     Rollout generator $\pi_\theta$ in multi-turn environment $\mathcal{E}$ to collect datapoints $\mathcal{D}_i \leftarrow \{(x, s_t, y_t, o_t))\}$
3:     Aggregate data $\mathcal{D} \leftarrow \mathcal{D} \cup \mathcal{D}_i$
4:     Train a verifier $R_\phi^i(x, y)$ on $\mathcal{D}$
5:     Construct a local search expert using verifier $\pi_\star^i(x) = \arg\max_{y \in \mathcal{D}(x)} \beta_O R(x, y) + \beta_L R_\phi(x, y)$
6:     Relabel data $\mathcal{D}$ with $\pi_\star^i(x)$ to get $\mathcal{D}_\star^i$
7:     Train $\pi_\theta^i$ with fine-tuning (FT) on $\mathcal{D}_\star^i$
8: **end for**

**output** Best generator $\pi_\theta$ and verifier $R_\phi$

---

## 3. $\mu$CODE: Multi-turn Code Generation

We propose $\mu$CODE, a simple and scalable algorithm for multi-turn code generation using execution feedback. $\mu$CODE follows an *expert iteration* (Anthony et al., 2017) framework with a *local search expert*. $\mu$CODE iteratively trains two components – a *learned verifier* $R_\phi$ to score code snippets (Section 3.2), and a *generator* $\pi_\theta$ to imitate local search with the verifier (Section 3.3). This iterative process allows the generator and expert to bootstrap off each other, leading to continuous improvement. At inference time, both the generator and verifier are used as BoN search to select and execute code (Section 3.4). Finally, we analyze the performance of $\mu$CODE in Section 3.5.

### 3.1. The $\mu$CODE Algorithm

Algorithm 1 presents the iterative training procedure. At an iteration $i$, the current generator $\pi_\theta$ is rolled out in the multi-turn code environment $\mathcal{E}$ to generate interaction data $\mathcal{D}_i \leftarrow \{(x, s_t, y_t, r_t)\}$. Every turn $t$ in $\mathcal{D}_i$ includes the prompt $x$, interaction history $s_t$, code generated $y_t$ and the correctness score from the oracle verifier $r_t = R(x, y_t)$. This data is then aggregated $\mathcal{D} \leftarrow \mathcal{D} \cup \mathcal{D}_i$. The learned verifier $R_\phi^i$ is trained on the aggregated data $\mathcal{D}$. An expert is created using $R_\phi^i$ to perform local search to find the optimal action $\pi_\star^i(x) = \arg\max_{y \in \mathcal{D}(x)} R_\phi^i(x, y)$, where $\mathcal{D}(x)$ are all the code completions for a given prompt $x$. The expert $\pi_\star^i(x)$ relabels the data $\mathcal{D}$ with the optimal action. The generator $\pi_\theta^i$ is then trained via fine-tuning (FT) on $\mathcal{D}$. This process iterates $M$ times, and the best generator and verifier pair on the validation dataset are returned.

### 3.2. Training Verifier

The learned verifier provides dense scores to code solutions for a given problem. At train time, this is used by the expert to perform local search to obtain optimal code. At inference time, the verifier is used for multi-turn BoN (3.4) for efficient search. The learned verifier has two distinct advantages over process reward functions typically used in multi-turn RL: (1) It is conditioned only on the initial prompt and the current solution, and is not dependent on previous states (2) It is trained via supervised learning on oracle reward labels. We explore two different losses:

**Binary Cross-Entropy loss** (BCE): The nominal way to train the verifier is to directly predict the oracle reward labels (Cobbe et al., 2021) as given by:

$$\mathcal{L}_{\text{BCE}}(\phi) = -\mathbb{E}_{(x,y,r)\sim\mathcal{D}}[r \log R_\phi(x, y) \\ -(1-r)\log R_\phi(x,y)] \qquad (2)$$

**Bradley Terry Model** (BT): Since the goal of the verifier is to relatively rank code solutions rather than predict absolute reward, we create a preference dataset and then train with a Bradley Terry loss (Ouyang et al., 2022). For every prompt $x$, we create pairs of correct $y^+$ (where $r = 1$) and incorrect $y^-$ (where $r = 0$) code and define the following loss:

$$\mathcal{L}_{BT}(\phi) = -\mathbb{E}_{(x,y^+,y^-)\sim\mathcal{D}}[\log \sigma(R_\phi(x, y^+) - R_\phi(x, y^-))]. \qquad (3)$$

where $\sigma(.)$ is the sigmoid function. We hypothesize that BT is strictly easier to optimize as the verifier has to only focus on relative performance. This is also consistent with observations made for training process reward models, where the advantage function is easier to optimize than the absolute Q function (Setlur et al., 2024).

### 3.3. Training Generator

$\mu$CODE comprises a generator $\pi_\theta$ trained to produce code solutions conditioned on the initial problem and execution observations from previous turns. Given a dataset $\mathcal{D}$, $\mu$CODE iteratively trains the generator to find the optimal code solution labeled using the local expert over the learned verifier. For this step, $\mu$CODE extracts all code solutions from $\mathcal{D}$ for every problem $x$. An expert is then created by picking the best solution, $y^\star$, which achieves the highest score using the learned verifier $R_\phi(x, y)$ when combined with the output of oracle verifier $R(x, y)$ and is given by

$$y^\star = \pi_\star(x) = \arg\max_{y \in \mathcal{D}(x)} \beta_O R(x, y) + \beta_L R_\phi(x, y), \quad (4)$$

where $\beta_O = 1.0$ and $\beta_L = 0.1$ denote the weights for oracle and learned rewards. Note that for creating the training dataset, we also use the ground labels from oracle verifier as they are available to the agent. The combination of both verifiers yielded better performance in our experiments. Using this expert dataset, we relabel $\mathcal{D}$ with the optimal solutions:

$$\mathcal{D}_\star = \{(x, s_t, y^\star) \mid (x, s_t) \sim \mathcal{D}\}, \qquad (5)$$

where $\mathcal{D}_\star$ represents the expert dataset. The generator $\pi_\theta$ is then trained via fine-tuning (FT) on this expert dataset $\mathcal{D}_\star$.

**Algorithm 2** $\mu$CODE: Inference loop

**input** Generator $\pi_\theta$, learned verifier $R_\phi$, turn limit T, number of rollouts N, public tests, and private tests
1: Set $s_1 = \{x\}$, $t = 1$
2: **while** true **do**
3:    Generate N rollouts $\{y_t^n\}_{n=1}^N \sim \pi_\theta(.|s_t)$
4:    Choose best solution $y_t^* = \arg\max_n R_\phi(x, y_t^n)$
5:    Execute $y_t^*$ to get execution feedback $o_t$
6:    **if** $y_t^*$ passes public tests or $t = T$ **then**
7:      break;
8:    **end if**
9:    Update state $s_{t+1} = \{s_t, y_t^*, o_t\}$ and increment $t$
10: **end while**
**output** Return $y^*$ to execute on public and private tests

### 3.4. Inference: Multi-turn Best-of-N

At inference time, the goal is to generate a code solution with a fixed inference budget – denoting the number of times generators can provide one complete solution. In this work, we propose to leverage the learned verifier to improve search and code generations over successive turns with *multi-turn Best-of-N* (BoN). To achieve this, $\mu$CODE uses a natural extension of BoN to the multi-turn setting. At each turn, the generator produces $N$ one-step rollouts $\{y_t^n\}_{n=1}^N \sim \pi_\theta(.|s_t)$ and the learned verifier picks the most promising code solution among these candidates using

$$y_t^* = \arg\max_n R_\phi(x, y_t^n). \quad (6)$$

The selected code $y_t^*$ is executed in the environment over public tests to obtain the execution feedback $o_t$. This solution and the feedback is provided as context to the generator at the next turn to repeat this procedure. The search ends once $y_t^*$ passes all public tests or when the turn limit is reached. Consequently, even if $R_\phi(\cdot)$ grants a high score to a code solution, inference continues until the solution has successfully cleared all public tests, thus mitigating potential errors by $R_\phi(\cdot)$. The final response $y_t^*$ is then passed through the oracle verifier to check its correctness. Algorithm 2 describes our propose procedure of multi-turn BoN search. We found it beneficial to use the reward model trained with samples of the latest generator $\pi_\theta$ (see Table 1).

### 3.5. Analysis

$\mu$CODE effectively treats multi-turn code generation as an interactive imitation learning problem by collecting rollouts from a learned policy and re-labeling them with an expert. It circumvents the exploration burden of generic reinforcement learning which has exponentially higher sample complexity (Sun et al., 2017). We briefly analyze why this problem is amenable to imitation learning and prove performance bounds for $\mu$CODE.

**Definition 3.1** (One-Step Recoverable MDP). A MDP $\mathcal{M} = (\mathcal{S}, \mathcal{A}, P, R, \gamma)$ with horizon $T$ is *one-step recoverable* if the advantage function of the optimal policy $\pi^*$, defined as $A^*(s, a) = Q^*(s, a) - V^*(s)$, is uniformly bounded for all $(s, a)$, i.e. $-1 \leq A^*(s, a) \leq 0$.

**Code generation is one-step recoverable MDP.** Multi-turn code generation satisfies one-step recoverability because the optimal policy $\pi^*(y_t|s_t)$ depends only on the problem prompt $x$ and not the interaction history $s_t = (x, y_1, o_1, \ldots, y_{t-1}, o_{t-1})$. Since the correctness of a code snippet $y_t$ is fully determined by $x$, the optimal Q-function satisfies $Q^*(s_t, y_t) = R(x, y_t)$, where $R(x, y_t) \in \{0, 1\}$. The optimal value function is $V^*(s_t) = \max_{y_t} R(x, y_t)$, so the advantage function simplifies to $A^*(s_t, y_t) = R(x, y_t) - \max_{y_t'} R(x, y_t') \leq 0$.

**Code generation enables efficient imitation learning.** There are two challenges to applying interactive imitation learning (Ross et al., 2011; Ross & Bagnell, 2014) – (1) Existence of expert policies or value functions, and (2) Recoverability of expert from arbitrary states. First, for code generation, the expert is simply the one-step reward maximizer $\arg\max_y R(x, y)$. We can efficiently estimate $R_\phi(x, y)$ to compute the expert, without needing to compute value function backups. Second, even if the learner fails to imitate the expert at any given state, the expert can perfectly recover from the next state. This results in the best possible performance bounds for imitation learning, which we formalize below.

**Theorem 3.2** (Performance bound for $\mu$CODE). *For a one-step recoverable MDP $\mathcal{M}$ with horizon $T$, running $N$ iterations of $\mu$CODE yields at least one policy $\pi$ such that*

$$J(\pi^*) - J(\pi) \leq O(T(\epsilon + \gamma(N))). \quad (7)$$

*where $\pi^*$ is the expert policy, $\epsilon$ is the realizability error, and $\gamma(N)$ is the average regret.*

Proof is in Appendix A.1. The bound $O(\epsilon T)$ is much better than the worst-case scenario of $O(\epsilon T^2)$ for unrecoverable MDPs (Swamy et al., 2021). Thus, $\mu$CODE exploits the structure of multi-turn code generation to enable imitation learning, bypassing the need for hierarchical credit assignment. More generally, this analysis suggests that for any task where the optimal action is history-independent and recoverable in one step, reinforcement learning can be reduced to efficient imitation learning without loss of performance.

## 4. Experiments

Through our experiments, we aim to analyze (1) How does $\mu$CODE compare to leading state-of-the-art methods? (2) Does a learned verifier facilitate training a better generator? (3) Can the use of a learned verifier improve multi-turn BoN

search at inference time? (4) Does the test-time search show scaling law trends? (5) Which loss function works better for learning a verifier for $\mu$CODE?

### 4.1. Setup

**Models.** The generator model in $\mu$CODE is initialized with Llama-3.2-1B-Instruct or Llama-3.1-8B-Instruct (Dubey et al., 2024). The learned verifiers are initialized with the same models as generators and have a randomly initialized linear layer to predict a scalar score (Stiennon et al., 2020).

**Datasets.** We conduct experiments on MBPP (Austin et al., 2021) and HumanEval (Chen et al., 2021) where the agent needs to generate code solutions in Python given natural language descriptions. We train the methods on the MBPP training set which comprises 374 problems and evaluate on the MBPP test set and HumanEval (HE) dataset which have 500 and 164 problems. We also compare methods on the DeepMind CodeContests dataset (CC, Li et al. (2022a)) where we train on 1000 problems sampled from the training set and evaluate on the 165 problems in the test set. We further describe the prompts and the split of public and private tests in Appendix C.1 and C.2. For training, we trained RFT and $\mu$CODE for 2 iterations in MBPP and HumanEval datasets and for 1 iteration on CodeContests.

**Baselines.** We compare $\mu$CODE with single and multi-turn baselines. For single and multi-turn settings, we report metrics with *Llama-3.2-1B Instruct* and *Llama-3.1-8B-Instruct* as base models. We also compare with rejection finetuning (RFT) where we collect multiple rollouts and filter trajectories with a correct solution for finetuning (FT). For multi-turn RFT, given a positive rollout $\{x, y_1, o_1, ..., y_T, o_T\}$, the model is finetuned over all the sub-trajectories $\{s_i, y_{i+1}\}_{i=0}^{T-1}$. For the multi-turn BoN search at inference time, we used the verifier learned from the last iteration for $\mu$CODE and trained a verifier with generated rollouts for the base and RFT models. Lastly, for multi-turn BoN search at inference time, we pick the best code solution with a hybrid approach where a solution passing public tests is preferred followed by ranking solutions with a learned verifier.

**Metrics.** We evaluate the methods by comparing the *BoN* accuracy. The generator is allowed upto $T = 3$ turns and the final solution is used for evaluation over private tests. The *BoN@1* evaluates the agent at producing the solution in the first attempt and is obtained via greedy decoding with a temperature of $0$. The *BoN* accuracy measures the ability of verifiers to leverage test-time compute by generating $N$ candidate solutions in parallel at each turn. At each turn, the verifier ranks $N = 5$ solutions (unless stated otherwise) provided by the generator. For the BoN performance, we

| Method | N | Llama-3.2-1B | | Llama-3.1-8B | | |
| | | *MBPP* | *HE* | *MBPP* | *HE* | *CC* |
|---|---|---|---|---|---|---|
| *Single-Turn* | | | | | | |
| Instruct | 1 | 35.1 | 25.6 | 52.1 | 59.8 | 3.6 |
| RFT | 1 | 35.7 | 34.1 | 53.7 | 54.9 | – |
| *Multi-Turn* | | | | | | |
| Instruct | 1 | 35.1 | 31.1 | 60.3 | 59.7 | 4.8 |
| +*BoN* | 5 | 47.3 | 35.7 | 69.7 | 62.9 | 13.8 |
| RFT | 1 | 31.1 | 31.7 | 58.9 | 61.2 | 7.2 |
| +*BoN* | 5 | 46.7 | 34.1 | 68.4 | 62.8 | 14.9 |
| $\mu$CODE | 1 | 37.9 | 35.4 | 62.1 | 60.9 | 7.9 |
| +*BoN* | 5 | **51.1** | **41.5** | **70.6** | **63.8** | **16.3** |

*Table 1.* Comparison of our method $\mu$CODE with baselines across MBPP, HumanEval, and CodeContests datasets. $N = 1$ denotes generating solutions with 0 temperature. The Best-of-N (BoN) accuracy is computed with $N = 5$ candidate solutions at each where the public tests and learned verifier are used for selection. We observe that $\mu$CODE outperforms competing methods based on Llama-3.2-1B-Instruct and Llama-3.1-8B-Instruct models. The best performance for each dataset and model-sized is highlighted in **bold** and similar performances (within 1%) are underlined.

sample with a temperature of $0.7$.

### 4.2. Results

In Table 1, we compare our proposed algorithm $\mu$CODE with the baselines. We first evaluate the generators using code generated via greedy sampling for each problem (*BoN* with $N = 1$). Our approach $\mu$CODE outperforms RFT across both benchmarks with 1B-sized model demonstrating the efficacy of using one-step recoverability and learned verifiers. To highlight, our method $\mu$CODE with the 1B model outperforms baselines by **2.2%** and **1.3%** on MBPP and HumanEval datasets. Interestingly, the performance of RFT drops when compared to the base Instruct model for the multi-turn setting which can be attributed to the fact that finetuning dataset consists of sub-trajectories with incorrect code solution at non-terminal steps. For the 8B-sized variant, we observe similar trends where we see that all algorithms benefit from the multi-turn BoN search at inference time. Additionally, $\mu$CODE performs better than both single and multi turn baselines across benchmarks.

With more inference budget and multi-turn BoN search at test-time, where the learned verifier and outcomes of public tests are used to select the best solution at each turn, we observe that every method performs better with the test-time search procedure by upto **13%**. For the 1B-sized models, our method $\mu$CODE significantly outperforms baselines by **4.4%** on MBPP and by **5.8%** on Humaneval datasets. For the 8B-sized variant, $\mu$CODE performs better than the

baselines across datasets. On the more challenging task of CodeContests, all methods benefit from the BoN search and observe performance gains of upto $2\times$. On this benchmark, $\mu$CODE outperforms the RFT and base model baselines by **1.4%**. We further investigate these trends with a Qwen model in Appendix B.1 and additional unit tests on MBPP and HumanEval benchmarks in Appendix B.2.

### 4.3. Analysis

To delve deeper into the improvements, we conduct a component-wise ablation study where we 1) check the effect of one-step recoverability and the learned verifier for creating the local expert (4.3.1), 2) compare different verifiers for multi-turn BoN search at test-time (4.3.2), 3) test the efficacy of agents at utilizing execution feedback (4.3.3), 4) assess scaling behaviors at inference time with number of candidate generations ($N$) at each turn (4.3.4), and 5) study different loss functions to train the verifiers (4.3.5).

#### 4.3.1. WHAT MAKES A GOOD GENERATOR IN $\mu$CODE?

| Method | One-step | Verifier | *MBPP* | *HE* |
|---|---|---|---|---|
| RFT | ✗ | Oracle | 46.7 | 36.5 |
| RFT$_{LV}$ | ✗ | Learned | 49.0 | 38.9 |
| $\mu$CODE$_{OV}$ | ✓ | Oracle | 48.2 | 38.0 |
| $\mu$CODE$_{LV}$ | ✓ | Learned | 48.5 | 39.1 |
| $\mu$CODE | ✓ | Both | **51.1** | **41.5** |

*Table 2.* Comparison of using learned verifier (LV) and relabeling with one-step recoverability (One-Step) with the 1B-sized model. We observe that FT with the learned verifier RFT$_{LV}$ performs better than FT with the oracle verifier scores RFT. Moreover, $\mu$CODE performs best when from both verifiers are used for relabeling.

Firstly, we compare different verifiers for training to demonstrate the benefits of using dense rewards obtained from learned verifiers. The RFT baseline uses the oracle verifier to filter trajectories and does not relabel subsequences for FT. In this experiment, we compare RFT to RFT$_{LV}$ where the learned verifier selects the top $K$ rollouts for each prompt ranked via the learned verifier for FT. We use $K = 3$ in our experiments. We observe in Table 2 that finetuning with the learned verifier outperforms the RFT baseline. In contrast to RFT, which lacks training data for prompts with no correct solutions, RFT$_{LV}$ effectively trains the generator to ascend the dense rewards obtained from the learned verifier.

Secondly, we present the advantages of relabeling subtrajectories with our insight of one-step recoverability. We extend the baselines with our proposed relabeling strategy described in Sec. 3.3. We call these methods $\mu$CODE$_{OV}$ ($\beta_O = 1$, $\beta_L = 0$) and CODE$_{LV}$ ($\beta_O = 0$, $\beta_L = 1$) depending on the verifier used for relabeling. Table 2 shows that $\mu$CODE$_{OV}$ outperforms the baseline RFT presenting the

| Approach | Llama-3.2-1B | | Llama-3.1-8B | |
|---|---|---|---|---|
| | *MBPP* | *HE* | *MBPP* | *HE* |
| **Base** | | | | |
| Random | 31.7 | 24.6 | 58.0 | 57.9 |
| LV | 30.3 | 29.4 | 62.5 | 61.0 |
| PT | 46.4 | 33.4 | 68.4 | 60.7 |
| PT+LV | 47.3 | 35.7 | 69.7 | 62.9 |
| **RFT** | | | | |
| Random | 31.1 | 27.4 | 58.9 | 57.7 |
| LV | 33.2 | 29.6 | 62.2 | 61.3 |
| PT | 46.8 | 36.8 | 67.4 | 61.4 |
| PT+LV | 46.7 | 36.5 | 68.4 | 62.8 |
| $\mu$**CODE** | | | | |
| Random | 37.5 | 31.5 | 61.5 | 58.4 |
| LV | 43.3 | 36.5 | 64.8 | 60.6 |
| PT | 49.4 | 39.7 | 69.4 | 61.4 |
| PT+LV | **51.1** | **41.5** | **70.6** | **63.8** |

*Table 3.* Comparing BoN with different ways of picking solutions at each turn for multi-turn BoN search using the 1B sized model. The hierarchical approach of using public test and learned verifier (PT+LV) outperforms picking solutions with only using either public tests (PT) or the learned verifier (LV). The best performance for each dataset and model-size is highlighted in **bold** and similar performances (within 1%) are underlined.

benefits of relabeling. The performance is similar for the setting where the learned verifier is used. To leverage the best of both worlds, $\mu$CODE uses a linear combination of scores from learned and oracle verifier for relabeling which outperforms each variant by around **2%**.

#### 4.3.2. DOES LEARNED VERIFIER AID BON SEARCH?

We study the effect of different verifiers for ranking the candidate solutions to pick the best solution for multi-turn BoN search at inference time. We test with *Random* strategy where the policy randomly picks from the $N$ solutions. We compare to using the outcomes on public tests (PT) that picks any solution that passes the public test. Note that this involves evaluating all generated solutions at every turn with all the given public tests. We also compare to selecting a solution based on scores obtained via the learned verifier only (LV). This is crucial as in certain applications such privileged information like public tests are not available and the agents can benefit from learned verifiers to improve search during inference. Lastly, we compare with the combination of public tests and using the dense scores obtained from the learned verifier to break ties at each turn (PT+LV).

In Table 3, we compare the baselines and $\mu$CODE with different verifiers at inference-time. We observe that LV outperforms *Random* strategy which shows that a learned verifier

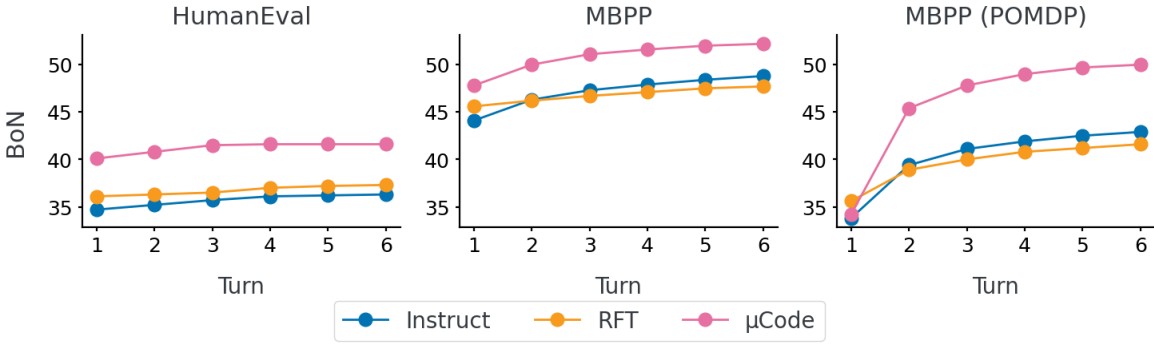

Figure 2. Compares the BoN performance at each turn (with 6 turns) on HumanEval, MBPP datasets. We also present results on a partially observable version of MBPP where we remove the public tests from the prompt to test the efficacy of methods at incorporating execution feedback– MBPP (POMDP). We observe that all methods improve performance with more turns on MBPP and HumanEval datasets. However, on MBPP (POMDP) the performance drops at first turn and $\mu$CODE closes the gap with performance on MBPP compared to the baselines demonstrating its ability to incorporate execution feedback to improve code solutions at each turn.

indeed selects better solutions amongst the candidates. Furthermore, using the outcome of public tests (PT) performed better than using learned verifiers (LV) and performs similarly on the HumanEval datset for 8B-sized models. We believe that this gap can be further reduced by learning more powerful verifiers with larger datasets. Interestingly, the hierarchical approach (PT+LV) that uses the learned verifier to break ties on the outcomes of public tests performs best across methods and datasets. We hypothesize that using learned verifiers is beneficial in two scenarios. Firstly, if multiple solutions pass the public tests, then the learned verifier can filter out incorrect solutions which may not pass private tests. Secondly, if all candidate solutions are incorrect, then the learned verifier chooses the most promising solution at each turn. This is crucial as picking a better solution with the learned verifier can lead to more relevant feedback for recovering the true solution.

### 4.3.3. CAN $\mu$CODE UTILIZE EXECUTION FEEDBACK?

We evaluate the ability of the trained generator to utilize execution feedback and improve the code response across turns. We report the BoN accuracy till a turn $t$, which denotes the accuracy of obtaining a correct solution within $t$ turns. Note that the agents were trained with rollouts of upto 3 turns and we report results with 6 turns for this experiment. In Fig. 2 (left and middle), we present the results with 1B-sized models where we observe that BoN accuracy improves with successive turns across the benchmarks.

To further understand if $\mu$CODE learns to utilize execution feedback, we curated another test dataset from MBPP where we removed the sample unit tests provided in the prompt, and call it MBPP (POMDP). Removing public unit test information from the prompt makes the information from execution feedback essential and is similar to a partially observable MDP setting. In Fig. 2 (right), we compare the

BoN accuracy over this evaluation dataset for $\mu$CODE and the baselines. We observe that the performance at first turn drops for all methods by upto **13%**. With the execution feedback at each turn, $\mu$CODE improves the code solutions leading to gains of **15.9%** from turn 1 to turn 6 and matches its performance on the MBPP dataset. In contrast, the base model and RFT were unable to close this gap in this partially observable setting and performed worse by around **6%** that the MBPP dataset. This demonstrates the ability of $\mu$CODE to recover better code solutions at each turn by utilizing the execution feedback at each turn, a trend not observed for the Instruct model and the RFT baselines.

### 4.3.4. DOES $\mu$CODE SCALE WITH INFERENCE BUDGET?

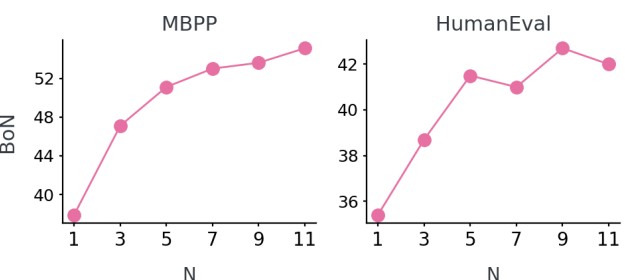

Figure 3. Test-time scaling with different values of candidate solutions $N$ at each turn. The candidate solutions are obtained from the 1B-sized generator of $\mu$CODE. We observe that the BoN performance improves with larger values of N on both datasets.

In the multi-turn setting, the number of candidate solutions can rise exponentially with the number of turns. To avoid this, $\mu$CODE uses the learned verifier during inference to select the most promising candidate among $N$ candidates at each turn, leading to a linearly increasing number of calls to the generator. In this experiment, we study the inference-time scaling behaviors of $\mu$CODE where we scale the num-

ber of candidate generations $N$ at each turn. Figure 3 plots the BoN with different values of $N$ ($1 \leq N \leq 11$). With more inference time budget, we observe that the performance improves with larger number of candidates at each turn on both datasets. The BoN accuracy plateaus with $N \geq 5$ for HumanEval dataset where for MBPP dataset we still observe some performance gains with larger $N$.

### 4.3.5. LOSS FUNCTION FOR VERIFIER

As described in 3.2, we compare against different loss functions for training the verifier. For this experiment, we first generate multiple single step rollouts and label them via oracle verifier. Given oracle labels, we train verifiers with two loss functions – BCE and BT. During inference, the learned verifier picks the best ranked solution among the $N$ solutions provided by the generator. Similar to (Cobbe et al., 2021), we report the BoN plot with different values of N obtained by first sampling $N$ candidate solutions, choosing the top-ranked solution using the learned verifier, and then evaluating the solution against public and private tests. We calculate this metric over multiple samples for each value of $N$. In Figure 4, we observe that the verifier trained with BT loss consistently outperforms the verifier trained on BCE loss on both MBPP and HumanEval.

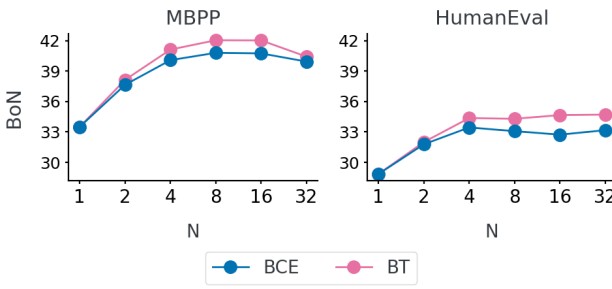

*Figure 4.* Comparison between BCE and BT loss function for training the verifier. We train the verifiers on samples generated by the base model (Llama-3.2-1B-Instruct). The learned verifier then ranks the candidate solutions from base model and the BoN performance of selected solution is reported. The verifier trained with BT loss performs better increasing value of N.

### 4.3.6. QUALITATIVE RESULT

Figure 5 presents a qualitative example of multi-turn Best-of-N search with $\mu$CODE. Through this example, we demonstrate the advantages of dense scores from the learned verifier at facilitating efficient search across turns. We generate $N = 5$ code solutions at each turn and show the top 3 ranked solutions using the dense scores. At the first turn, we observe that the last solution $y_1^3$ is less accurate than the other 2 solutions $y_1^1$ and $y_1^2$. The top ranked solution is used to collect the environment feedback, upon which the generator comes up with $N$ new candidate solutions. Upon

the top 3 solutions, the last two snippets are similar to the candidates from the previous turn. However, the top ranked solution is a novel solution and is more accurate as the generated code learns to extract a single digit and multiply it. With the execution feedback, $\mu$CODE generates 2 correct responses– $y_3^1$ and $y_3^2$ and learned verifier chooses one of them compared to the incorrect response $y_3^3$.

## 5. Related Work

**Prompting To Solve Multi Step Tasks**   A common framework for tackling multi-step tasks with LLMs is prompting-based agentic systems. Self-Debugging (Chen et al., 2023b) asks the LLM to iteratively improve code by providing execution feedback while CodeT (Chen et al., 2022) asks the LLM to generate test cases. AlphaCodium (Ridnik et al., 2024) first reflects on input instructions, generates and filters from multiple code generations, and finally iterates on public and self-generated test cases. MapCoder (Islam et al., 2024) incorporates four agents to generate example problems, plans and code, and then perform debugging. However, prompting-based agents yield limited improvements.

**Training LLMs for Multi Step Tasks**   Some work has explored explicitly training critics or reward models for multi-step reasoning tasks. In the coding domain, CodeRL (Le et al., 2022) trains a token-level critic to aid in code generation and to perform inference-time search. CodeRL's mechanics are similar to our method, but their generator is not trained for multi-step: CodeRL trains a "code repairer" which conditions on one erroneous code completion while our generator incorporates multiple. ARCHER (Zhou et al., 2024), which frames multi-step tasks via a two-level hierarchical MDP, where the higher level MDP considers completions as actions and the lower level MDP considers tokens as actions. Another line of work utilizes Monte Carlo Tree Search (MCTS) methods for training: rStar-Math (Guan et al., 2025) trains a policy preference model to boost small LMs' math abilities to match or exceed large reasoning-based LMs and ReST-MCTS (Zhang et al., 2024) trains a process reward model (PRM) similarly to Math-Shepherd (Wang et al., 2024a). Although $\mu$CODE's BoN search resembles a tree search, our key insight that multi-step code generation resembles a one-step recoverable MDP allows us to collect training trajectories much more efficiently. Finally, some work has explored using verifiers only during inference time. In "Let's Verify Step by Step" (Lightman et al., 2023), the authors demonstrate that PRMs trained on erroneous math solutions annotated by humans outperform outcome reward models for filtering multiple inference time generations. Meanwhile, AlphaCode (Li et al., 2022b) trains a test generator to evaluate multiple code solutions.

Other works omit learning a critic or reward model alto-

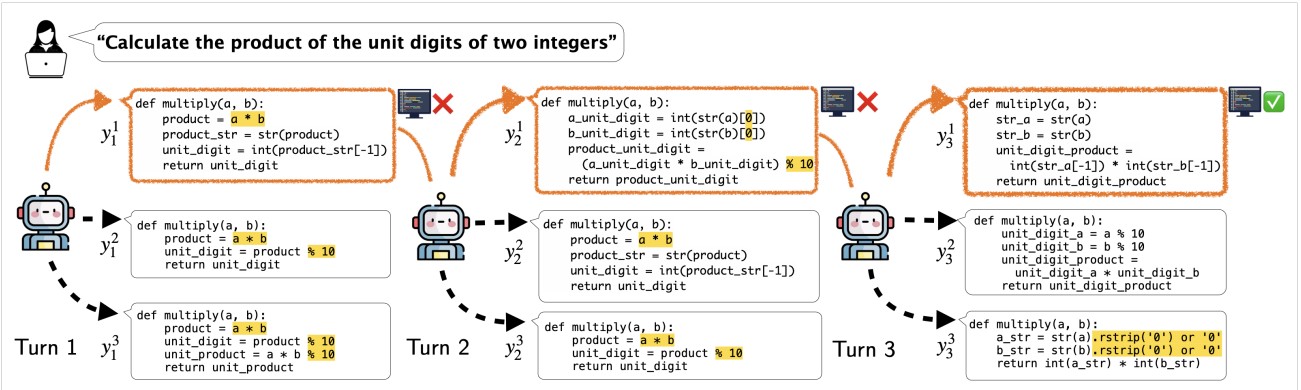

*Figure 5.* A qualitative example of multi-turn BoN search using dense rewards obtained via the learned verifier in $\mu$CODE. Here, we show the top 3 ranked solutions at each turn $t$ where $R_\phi(x, y_t^i) \geq R_\phi(x, y_t^j)$ for $i < j$. We observe that the learned verifier selects the better solution (in orange) at each turn. The selected solution is passed to public tests to retrieve execution feedback for the generator to improve the next code solution. The selected solution at each turn is better than the last (less errors highlighted in yellow), with the final solution passing all tests. Note that there are 2 correct solutions at the final turn.

gether. In the coding domain, RLEF ([Gehring et al., 2024b](#)) derives rewards only on the executor's result on test cases and syntax checkers, and PPOCoder ([Shojaee et al., 2023](#)) additionally considers semantic and syntactic alignment, generated via data flow graphs and abstract syntax trees respectively, with a reference solution. The "oracle" rewards in these methods may not be informative for training, and in the case of PPOCoder, require complex constructs. We empirically show that having a reward model is beneficial by comparing $\mu$CODE against the RFT baseline. Meanwhile, SCoRe ([Kumar et al., 2024a](#)) splits training into a "generator" and "correction" phase, thus restricting the total number of turns to 2. RISE ([Qu et al., 2024](#)) generates recovery steps via a more powerful LLM or by selecting a sampled completion via the oracle rewards. Both methods are less efficient than $\mu$CODE, which doesn't require generating corrections beyond generating training trajectories. Finally, FireAct ([Chen et al., 2023a](#)) and LEAP ([Choudhury & Sodhi, 2024](#)) FT ReAct style agents while RL4VLM ([Zhai et al., 2024](#)) and GLAM ([Carta et al., 2024](#)) studies training LLMs with interactive environment feedback.

## 6. Conclusion

We present $\mu$CODE, a simple and scalable method for multi-turn code generation through single-step rewards. $\mu$CODE models code generation as a one-step recoverable MDP and learns to iteratively improve code with a learned verifier to guide the search. Experimental results demonstrate that $\mu$CODE outperforms methods using oracle verifiers by a large margin. We acknowledge the following limitations of this paper. Due to a limited budget, we were only able to train models with up to eight-billion parameters. It is possible that the conclusions made in this paper do not

generalize to models of larger scales. Additionally, we train models on MBPP, whose training set has only 374 examples. However, we hypothesize that more training examples will lead to better performance. Finally, our datasets are only in Python, and our findings might not generalize to other programming languages.

## Impact Statement

The proposed method for training code agents has the potential to streamline software development processes by automating routine coding tasks, thereby reducing human labor and accelerating production timelines. However, these advances will also introduce bugs, which can propagate at scale if no proper quality control is in place.

## Acknowledgements

AJ is supported by Calcul Québec, Canada Excellence Research Chairs (CERC), and Fonds de Recherche du Québec (FRQ) scholarship (DOI assigned: https://doi.org/10.69777/350253) program. The authors are also grateful to Mila (mila.quebec) IDT and Digital Research Alliance of Canada for computing resources. AMR is supported in part by NSF CAREER #2037519 and NSF #2242302. SC is supported in part by Google Faculty Research Award, OpenAI SuperAlignment Grant, ONR Young Investigator Award, NSF RI #2312956, and NSF FRR#2327973.

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

# A. Proofs

## A.1. Proof of Theorem 3.2

The proof relies on two important results.

The first is the Performance Difference Lemma (PDL) (Kakade & Langford, 2002) which states that the performance difference between any two policies can be expressed as the sum of advantages.

$$J(\pi) - J(\pi') = \sum_{t=1}^{T} \mathbb{E}_{s_t \sim d_t^\pi} \left[ \sum_{a_t} A^{\pi'}(s_t, a_t) \pi(a_t|s_t) \right] \tag{8}$$

where $s_t \sim d_t^\pi$ is the induced state distribution by $\pi$, and $A^{\pi'}(s_t, a_t) = Q^{\pi'}(s_t, a_t) - V^{\pi'}(s_t)$ is the advantage w.r.t. $\pi'$.

We apply the PDL between the expert $\pi^*$ and the learner $\pi$

$$J(\pi^\star) - J(\pi) = \sum_{t=1}^{T} \mathbb{E}_{s_t \sim d_t^\pi} \left[ \sum_{a_t} A^\star(s_t, a_t) \left( \pi^\star(a_t|s_t) - \pi(a_t|s_t) \right) \right] \tag{9}$$

where the result follows from $\left( \sum_{a_t} A^\star(s_t, a_t) \pi^\star(a_t|s_t) = 0 \right)$

According to the one-step recoverable MDP definition, $-1 \leq A^\star(s,a) \leq 0$ for all $(s,a)$. Hence we can bound the performance difference as

$$
\begin{aligned}
J(\pi^\star) - J(\pi) &= \sum_{t=1}^{T} \mathbb{E}_{s_t \sim d_t^\pi} \left[ \sum_{a_t} A^\star(s_t, a_t) \left( \pi^\star(a|s_t) - \pi(a|s_t) \right) \right] \\
&\leq ||A^\star(.,.)||_\infty \sum_{t=1}^{T} \mathbb{E}_{s_t \sim d_t^\pi} ||\pi(.|h_t) - \pi^\star(.|s_t)||_1 \qquad \text{(Holder's Inequality)} \\
&\leq \sum_{t=1}^{T} \mathbb{E}_{s_t \sim d_t^\pi} ||\pi(.|s_t) - \pi^\star(.|s_t)||_1 \qquad \text{(One step recoverability)}
\end{aligned}
$$

The second result we use us from interactive imitation learning DAGGER (Ross et al., 2011) that reduces imitation learning to no-regret online learning. DAGGER shows that with $\pi^\star$ as the expert teacher guarantees that after $N$ iterations, it will find at least one policy

$$\mathbb{E}_{s \sim d^\pi} ||\pi(.|s) - \pi^\star(.|s)||_1 \leq \mathbb{E}_{s \sim d^\pi} ||\pi_{\text{class}}(.|s) - \pi^\star(.|s)||_1 + \gamma(N) \tag{10}$$

where $\gamma(N)$ is the average regret, and $d^\pi$ is the time average distribution of states induced by policy $\pi$, $\pi_{\text{class}}$ is the best policy in policy class.

Plugging this in we have

$$
\begin{aligned}
J(\pi^\star) - J(\pi) &\leq \sum_{t=1}^{T} \mathbb{E}_{s_t \sim d_t^\pi} ||\pi(.|s_t) - \pi^\star(.|s_t)||_1 \\
&\leq \sum_{t=1}^{T} \mathbb{E}_{s_t \sim d_t^\pi} ||\pi_{\text{class}}(.|s_t) - \pi^\star(.|s_t)||_1 + \gamma(N) \quad \text{From (10)} \\
&\leq T(\epsilon + \gamma(N))
\end{aligned}
$$

# B. Additional Results

## B.1. Qwen Model

| Method | N | *MBPP* | *HE* |
|--------|---|--------|------|
| Instruct | 1 | 53.8 | 64.5 |
| *+BoN* | 5 | 60.9 | 70.3 |
| RFT | 1 | 57.0 | 66.6 |
| *+BoN* | 5 | 58.0 | 71.3 |
| $\mu$CODE | 1 | 59.0 | 70.5 |
| *+BoN* | 5 | **63.1** | **74.0** |

*Table 4.* Comparison of our method $\mu$CODE with baselines across MBPP, HumanEval using the Qwen-2.5-1.5B-Instruct model.

## B.2. Additional private tests

| Method | N | *MBPP+* | *HE+* |
|--------|---|---------|-------|
| Instruct | 1 | 43.2 | 25.7 |
| *+BoN* | 5 | 49.9 | 31.9 |
| RFT | 1 | 44.3 | 26.7 |
| *+BoN* | 5 | 50.0 | 34.3 |
| $\mu$CODE | 1 | 48.7 | 32.4 |
| *+BoN* | 5 | **55.1** | **40.0** |

*Table 5.* Comparison of our method $\mu$CODE with baselines across MBPP+, HumanEval+ using the Llama-3.2-1B-Instruct model. MBPP+ and HumanEval+ introduce 35x and 80x more tests than their original counterparts respectively. Note that MBPP+ has a higher score than MBPP because there are 30% fewer problems

# C. Hyperparameters

| Model | Generator | Verifier |
|-------|-----------|----------|
| Training Epochs | 2 | 2 |
| Learning Rate | $5 \times 10^{-7}$ | $1 \times 10^{-6}$ |
| Batch Size | 32 | 64 |
| Max seq length | 4096 | 2048 |

*Table 6.* Hyperparameters for SFT and RM training.

### C.0.1. TRAINING PARAMETERS

Table 6 contains hyperparameters for training the generator and reward model on both models (Llama-3.1-8B-Instruct and Llama-3.2-1B-Instruct) and datasets (MBPP and HumanEval). We perform 2 iterations of training with $\mu$CODE, starting from the base model each iteration. All training runs were on machines with either 4 RTX 6000 Ada Generation GPUs for 1B models with 48 GB of memory per GPU or 4 H100 GPUs for 8B models with 80 GB of memory per GPU.

### C.0.2. INFERENCE PARAMETERS

We use SGLang (Zheng et al., 2024) to serve our models for inference. Greedy experiments use temperature 0 with flags *–disable-radix-cache –max-running-request 1* to ensure deterministic results while BoN search experiments use a temperature of 0.7. All experiments are capped to 1000 tokens per completion per turn.

## C.1. Prompts

### C.1.1. SINGLE STEP PROMPT

Immediately below is the prompt template to generate 1 code completion in a single-step method or to generate the 1st step in a multi-step method. Below the prompt templates are examples of the code prompt and public tests for HumanEval, MBPP, and CodeContests.

---

**Single Step Prompt**

```
Write a Python function implementation for the following prompt:

\{prompt\}

Your code should satisfy these tests:

\{test\}

Please follow the following instructions:
- Reason about the problem and any base cases before writing the code.
- You must return the implementation code in the following format:
```python
<CODE GOES HERE>
```
- You must only return a single code block since we only parse the first code block.
- Do not include any tests in your code - we will run the suite and return any error
  feedback.
- Include relevant import statements.
```

---

**HumanEval Prompt Example**

```
from typing import List

def has_close_elements(numbers: List[float], threshold: float) -> bool:
    """ Check if in given list of numbers, are any two numbers closer to each other
than
    given threshold.
    >>> has_close_elements([1.0, 2.0, 3.0], 0.5)
    False
    >>> has_close_elements([1.0, 2.8, 3.0, 4.0, 5.0, 2.0], 0.3)
    True
    """
```

---

**HumanEval Test Example**

```
def check(has_close_elements):
    assert has_close_elements([1.0, 2.0, 3.0], 0.5) == False
    assert has_close_elements([1.0, 2.8, 3.0, 4.0, 5.0, 2.0], 0.3) == True
check(has_close_elements)
```

---

**MBPP Prompt Example**

```
Write a function to find the minimum cost path to reach (m, n) from (0, 0) for the
given cost matrix cost[][] and a position (m, n) in cost[][].
```

---

**MBPP Test Example**

```
assert min_cost([[1, 2, 3], [4, 8, 2], [1, 5, 3]], 2, 2) == 8
assert min_cost([[2, 3, 4], [5, 9, 3], [2, 6, 4]], 2, 2) == 12
assert min_cost([[3, 4, 5], [6, 10, 4], [3, 7, 5]], 2, 2) == 16
```

---

**CodeContests Prompt Example**

```
Provide a Python solution for the following competitive programming question:

Mr. Chanek has an array a of n integers. The prettiness value of a is denoted as:

$$$\Sigma_{i=1}^{n} {\Sigma_{j=1}^{n} {\gcd(a_i, a_j) \cdot \gcd(i, j)}}$$$

where \gcd(x, y) denotes the greatest common divisor (GCD) of integers x and y.

In other words, the prettiness value of an array a is the total sum of \gcd(a_i, a_j
) \cdot \gcd(i, j) for all pairs (i, j).

Help Mr. Chanek find the prettiness value of a, and output the result modulo 10^9 +
7!

Input

The first line contains an integer n (2 \leq n \leq 10^5).

The second line contains n integers a_1, a_2, ..., a_n (1 \leq a_i \leq 10^5).

Output

Output an integer denoting the prettiness value of a modulo 10^9 + 7.

Example

Input

5
3 6 2 1 4

Output

77

Your code should be enclosed in triple backticks like so: ```python YOUR CODE HERE
```. Use the backticks for your code only.
```

---

**CodeContests Test Example**

```
# Input fed through stdin and output checked against stdout
{'input': ['5\n54883 59286 71521 84428 60278\n', '2\n83160 83160\n'], 'output':
['1027150\n', '415800\n']}
```

---

### C.1.2. FEEDBACK PROMPT

Immediately below is the prompt template for how we provide feedback in multi-step methods. The feedback only consists of executor error traces, and we provide an example from HumanEval.

---

Multi-Step Feedback Prompt

```
Feedback:

\{feedback\}
```

---

HumanEval Multi-Step Feedback Prompt

```
Traceback (most recent call last):
  File "test.py", line 18, in <module>
    assert has_close_elements([1.0, 2.0, 3.0], 0.5) == False
           ^^^^^^^^^^^^^^^^^^^^^^^^^^^^^^^^^^^^^^^^^^^^^^^^^^^
AssertionError
```

---

## C.2. Public Private Tests

We choose a public-private test split for HumanEval and MBPP to ensure that naively passing the public tests does not guarantee private test success. For HumanEval, we use a single test from the code prompt's docstring as the public test and the remaining tests along with the official test suite as private tests. For ease of parsing, we utilize a processed version of HumanEval, HumanEvalPack (Muennighoff et al., 2023). For MBPP, we use a single test from the official test suite as the public test, and the remaining tests and any "challenge test list" tests as private tests.

