# OpenReview forum: "Multi-Turn Code Generation Through Single-Step Rewards"
_ICML.cc/2025/Conference — ICML 2025 spotlightposter_

### Official Review · Reviewer_6VTR · 2025-03-08

**Overall Recommendation:** 4

**Summary:**

This paper tackles the task of multi-turn (MT) code generation by introducing $\mu$-Code, an *expert iteration* process to alternate between training a generator (which produces candidate code solutions $y$ conditioned on past history $s$) and training a verifier ($R(x, y)$ which scores these solutions). This is motivated by the observation that MT code generation is a one-step recoverable MDP, as a correct solution could be produced conditioned on any history. Experiments of 3-turn code generation on 2 benchmarks, MBPP and HumanEval, show that it performs better than the MT baseline (fine-tuning on the correct MT trajectories): 1. training using a learned verifier performs better than using an oracle verifier ($\mu$-Code v.s. Multi-STaR$^2$ and Multi-STaR). 2. learned verifier could better guide the MT code generation process in the multi-turn BoN (conditioned on a history, do N rollout out, and select the highest score as the current prediction) process.

**Claims And Evidence:**

The claim that a learned verifier’s single-step reward is sufficient for multi-turn code generation is well motivated by the proposed one-step recoverable MDP formulation. The authors use the learned verifier to pick top candidates to relabel the dataset. They provide empirical improvements over STaR-like baselines (which use correct completions as fine-tuning data) in training time and show that the learned verifier could be combined with public test information to further boost the BoN performance. The authors also provide single-turn baselines in Table 1 for solid comparison.

However, the lack of experiment details/hyperparameters makes it difficult to assess whether the comparison is sound: I'm in particular concerned with the experiment results that:

1. The performance drops in general from Iter 1 to Iter 2 on MBPP for $\mu$-Code and on both MBPP and HumanEval for Multi-StaR$^2$. Therefore, it remains a question whether the proposed expert iteration process works for Iteration number >= 3.

2. The results in Table 2 where $\mu$-Code-R is worse than $\mu$-Code is counter-intuitive and more explanation from the authors on the reason behind or on the experiment setting is highly appreciated.

Also, as the authors discuss in the limitations, apart from limited resources for larger experiments, it remains unclear whether the claims could extend to:
1. other benchmarks beyond MBPP and HumanEval, essentially, where the public tests information is not fully presented in the prompt $x$, making the information from execution feedback essential (in a Partially-Observable MDP setting).
2. a larger expert iteration number (>= 3).

**Essential References Not Discussed:**

I encourage the authors to include https://arxiv.org/abs/2410.08105, in which the authors compare different chain-of-thought strategies in multi-turn code generation, and the described multi-turn Rejection Sampling Finetuning is also related to what the authors describe in L259 about the Multi-StaR.

**Experimental Designs Or Analyses:**

Questions about the experimental details remain for correctly assessing the experiment results:

1. Understanding the baseline setting: The StaR https://arxiv.org/pdf/2203.14465 describes 2 steps to construct a dataset for finetuning: 1. Filtering Chain-of-Thought based on the final correct answer. 2. Conditioned on the correct answer, let the model generate Chain-of-Thought. While what the authors described as StaR in the paper (L258-260) is only doing a variant of the Step 1 above, i.e., rejection sampling and fine-tuning. This is closer to what is referred to as Rejection Sampling Finetuning, i.e., rollout from the models and fine-tuning on the correct ones (in https://arxiv.org/abs/2307.09288 and https://arxiv.org/abs/2308.01825 for single-turn setting in code/math reasoning domain). The description of Multi-StaR (L260), i.e., fine-tuning on correct multi-turn rollout, is very similar to the multi-turn Rejection Sampling Finetuning in https://arxiv.org/abs/2410.08105. It remains a question about whether the authors' StaR/Multi-StaR setting is closer to the original StaR involving Chain-of-Tought re-ordering and boostrapping or closer to the above Rejection Sampling Finetuning.

2.  I'm in particular curious about the drop of performance from Iter 1 to Iter 2 in Table 1. Could the author provide any insight? In the training loop of the expert iteration L7 in Algo 1, new policy for iteration N is always fine-tuned from the base model (iteration 0), or it's iteratively fine-tuned from the policy of last iteration (iteration N - 1)?

3. In Eq(3), how are the correct/incorrect pairs constructed? Are you doing a full catesian product of correct x incorrect solutions or there're some sampling processes?

4. In Eq(5), do you do the relabelling for all the problems, even if there's no correct solutions to that problem?

5. What are the sizes of the datasets for each variants in Table 2 or other statistics of the dataset, e.g., distribution of trajectories with different turn number, distribution of the correct/incorrect trajectories in the relabelled dataset for $\mu$-Code? It remains unclear why $\mu$-Code-R is better than $\mu$-Code to me despite more correct trajectories are added.

**Methods And Evaluation Criteria:**

The authors train on MBPP training set and evaluate on MBPP test set and HumanEval. This is a valid choice given the size of the model that the authors experiment on (1B & 8B). The authors report pass@3 and BoN@3 to report multi-turn (here 3-turn) performance. The authors include single-turn performance in Table 1, which I found solid for strengthening the comparison.

**Other Comments Or Suggestions:**

Please see above.

**Other Strengths And Weaknesses:**

Overall, I have mixed feelings about the manuscript, on one hand the strengths are clear:
1. The one-step recoverable MDP formulation is interesting and intuitive.
2. Performance gain through extensive baseline comparison. Especially the Multi-StaR$^2$, which I see as the most important ablation of the whole proposed pipeline by replacing the learned verifier with the oracle verifier but still keeping the relabelling process. However, it could benefit from more discussion about the source of the performance gap. I only find text from L310-312 for this comparision.

On the other hand, I'm not fully convinced by some of the results, along with the following weaknesses, for which I would need more clarification from the authors to recommend an acceptance:
1. Questions remain regarding my comments on the theoretical part above.
2. A lot of experiment details are missing and would be great if the authors could clarify my questions above on the experimental designs part.
3. Reproducibility: It would be great at least to discuss the experiment setting, such as hyperparameters.

**Questions For Authors:**

Please see the above comments. Overall, I remain positive about the manuscript. I would consider revising my recommendation if the following concerns and confusing are addressed:

Major:
1. Theoretical and Experimental Design.
2. Intuition and more explanation on why $\mu$-Code-R is worse than $\mu$-Code.

Minor:
1. The other questions on the Claim part, namely the discussion on whether the claims could be extend to iteration >= 3 or POMDP.

**Relation To Broader Scientific Literature:**

- https://arxiv.org/pdf/2405.17503 maintains a search tree and uses Thompson Sampling to choose the multi-turn history to condition on in the multi-turn code generation.
- https://arxiv.org/abs/2410.08105 compares different chain-of-thought strategies in multi-turn code generation, and the described multi-turn Rejection Sampling Finetuning is also related to what the authors describe in L259 about the Multi-StaR.

**Theoretical Claims:**

The authors propose the one-step recoverable MDP and prove the performance bound. I have checked the proof in Appendix B. The proof seems correct in general, but I have some questions/comments regarding the definition and the details in the proof.

- The transition from L733 ($A^*(s, a) \le 1$) to what is used in L740 (I understand that the authors want to use $||A^*(.,.)||_{\infty} \le 1$). This is not that correct because, in principle, $A^*(s, a)$ could be negative and below -1, and the absolute value could be bigger than 1.

- In my opinion, the definition in L206: "MDP is one-step recoverable if the advantage function of the optimal policy, defined as $A^*(s, a) = Q^*(s, a)−V^*(s)$, is uniformly bounded for all (s, a), i.e. $A^*(s, a) \le 1$" is very artificial and is not really the definition. Rather, this should an implication from what the authors argue (from L211 to L213) that "MDP is one-step recoverable if the optimal policy depends only on the problem prompt $x$ but not the interaction history" and I would further argue that:

1. The definition is not correct in its current form.
2. The implication should have a tighter bound. But more importantly, it has nothing to do with the "one-step recoverability"; it's generally true and trivial for any MDP.

For my point 1, we can construct a toy MDP environment, which is according to the current definition and is not one-step recoverable MDP. We allow 4 states $s_0 (initial), s_1, s_2 (terminal), s_3 (terminal)$. There are only 2 possible deterministic transitions:
1. $s_0, a_0$ gives $s_2$, reward 1.
2. $s_0, a_1$ gives $s_1$, reward 0, and $s_1, a_2$ gives $s_3$, reward 2.

If this is one-step recoverability, the optimal policy should only depend on the initial state (L211) but not the history. However, the optimal policy here should pick the trajectory of $s_0, s_1, s_3$, and, therefore, is history-dependent. And we have (assuming here discount factor $\gamma$ is 1):

1. $A^*(s_0, a_0) = 1 - 2 = -1 \le 1$
2. $A^*(s_0, a_1) = 2 - 2 = 0 \le 1$
3. $A^*(s_1, a_2) = 2 - 2 = 0 \le 1$

which satisfies perfectly the current definition, which states that all $A^*$ should be smaller than or equal to 1.

For my point 2, $V^*(s) = max_a Q^*(s, a)$ by definition and it's direct that $A^*(s, a) = Q^*(s, a) - V^*(s) = Q^*(s, a) - max_a Q^*(s, a) <= 0 < 1$. Therefore, the claim that $A^*(s, a) \le 1$ is trivial, true for any MDP, and it's not even a good bound. Unless the authors want to say that it's negatively bounded as well in the current setting.

I think one way to fix all these is to *rewrite the definition*, **and** put $-1 \le A^*(s, a) \le 0$ (the -1 is environment specific, which is from the range of R = 0 or 1) in the definition instead of $A^*(s, a) \le 0$. In all, the definition still looks interesting if all my points above are properly addressed.

---

> ### Author Rebuttal · Authors · 2025-04-01
>
> We thank the reviewer for the valuable feedback. We are glad the reviewer found the idea of one-step recoverability interesting. We respond to the questions below.
>
> > Results on POMDP setting.
>
> This is an interesting ablation as for many prompts, some unit tests are present in the $x$. To evaluate the agents without this information in prompt, we removed the unit tests from the prompt $x$ in the test set of MBPP dataset and report the turn-wise BoN@3 score. We observe that the performance drops for all algorithms. We also observe that the performance gains for $\mu$Code is significantly higher than baselines across turns, demonstrating the efficacy of our proposed approach at leveraging execution feedback.
>
> | Algorithm              | Turn 1 | Turn 2 | Turn 3 |
> | :---------------- | :------: | :------: | :------: |
> | Llama-3.2-1B-Instruct        |24.3|29.4|32.6
> | Multi-STaR  |25.2|30.3|32.3
> | Multi-STaR^2|22.8|26.8|27.4
> | $\mu$Code   |**27.3**|**37.2**|**41.5**
>
> > Other benchmarks beyond MBPP and HumanEval
>
> We evaluate the methods on the CodeContests dataset and is discussed in the response to Reviewer QsKU. We show that $\mu$Code outperforms Multi-STaR by 1.6%.
>
> > The results in Table 2 where $\mu$-Code-R is worse than $\mu$Code.
>
> To generate the dataset for training $\mu$Code-R, we concatenate the datasets used to train $\mu$-Code and MultiSTaR^2. We believe a better way of merging might lead to reduce this gap in performance and plan to add an additional analysis in the next version.
>
> > Whether $\mu$Code works for Iteration number >= 3.
>
> In Table 1, we observe a performance drop with $\mu$Code on the MBPP dataset and observe improvements on HumanEval with training iterations. Given the simplicity of our approach, we believe more iterations would be needed for scaling to more challenging datasets and we leave this investigation for future research.
>
> > Theoretical claims
>
> We thank the reviewer for pointing this out. Using the reward function defined in Section 2 (L100) to take values of either 0 or 1, we have updated the paper with the $-1 \leq A*(s,a) \leq 0 $ in Definition 3.1, L218 and L733.
>
> > Whether StaR/Multi-StaR setting is closer to the original StaR  or Rejection Sampling Finetuning.
>
> The Multi-STaR baseline used in the paper is similar to the Rejection Sampling Finetuning defined in [2]. We plan to change the name of Multi-STaR to RFT for more clarity, and also change Multi-STaR^2 to RFT-R to denote RFT with relabeling.
>
> > Is the new policy for iteration N is always fine-tuned from the base model?
>
> In this work, we followed a similar strategy as [1] for fine-tuning the base model at every iteration.
>
> > In Eq(3), how are the correct/incorrect pairs constructed?
>
> The dataset for training the verifier was obtained through a sampling process. For each prompt, a pair of correct and incorrect response was obtained and we generated 16 pairs for each prompt.
>
> > In Eq(5), do you do the relabelling for all the problems?
>
> Yes, $\mu$Code relabels all the problems even if there are no correct solutions to that problem. This is similar to how RL training would happen with a dense reward function, where the agent is updated to increase the likelihood of the response with higher reward values and climb uphill on the reward function with updates. $\mu$Code relabels the trajectories with solutions with higher reward values for fine-tuning.
>
> > What are the sizes of the datasets for each variants in Table 2 or other statistics of the dataset?
>
> At each iteration, the size of the dataset for $\mu$Code is the total number of rollouts as we relabel every trajectory. For the Multi-STaR^2, we relabel ~70% of the rollouts denoting the fraction of prompts with at least one positive trajectory. The dataset for $\mu$Code-R is obtained by concatenating the two datasets. We ran hyperparameter tuning on the number of epochs to account for different sized datasets but finetuning with 2 epochs performed best.
>
> > Reproducibility:
>
> We will be releasing the codebase and the model checkpoints with the paper. We provide details of important hyperparameters for training $\mu$Code below:
> | Hyperparameter | Generator | Verifier |
> | :---------------- | :------: | :------: |
> | Training Epochs        |2|2|
> | Learning Rate          |5$e^{-7}$|1$e^{-6}$|
> | Batch Size             |32|64|
> | Max Sequence Length    |8192|2048|
>
>
> > Essential References Not Discussed
>
> [2] is a very interesting work that studies different CoT approaches and execution feedbacks. We have added [2] in references as the Multi-STaR baseline is similar to the Rejection Sampling Finetuning method described in [2].
>
> Best Regards,
>
> The Authors
>
> ### References
> [1]  Qu et al.,Recursive Introspection: Teaching Language Model Agents How to Self-Improve, NeurIPS 2024
>
> [2] Zheng et al., What Makes Large Language Models Reason in (Multi-Turn) Code Generation?,  ICLR 2025

---

> > ### Comment · Reviewer_6VTR · 2025-04-04
> >
> > I thank the authors for the response which resolves most of my concerns. The motivation and contributions of the paper is clear and weakness in my original reviews are addressed (the new POMDP and the CodeContests result). What remain unsolved within the rebuttal session are also easily fixable, e..g flaws in the therectical part, the wording/notation, and the essential exposure about hyperparameters.
> >
> > > To generate the dataset for training $\mu$Code-R, we concatenate the datasets used to train $\mu$-Code and MultiSTaR$^2$. We believe a better way of merging might lead to reduce this gap in performance and plan to add an additional analysis in the next version.
> >
> > This analysis important as intuitively, adding more data from oracle verifier (i.e., the environment) for fine-tuning should not hurt the perf.
> >
> > I encourage the authors to further polish the manuscript and include more details discussed here into the next version, as well as what the authors promise about the addditional analysis. As such I raise my score and recommend an acceptance.

---

> > > ### Author Response · Authors · 2025-04-08
> > >
> > > Dear Reviewer 6VTR,
> > >
> > > We would like to sincerely thank you for your support and are glad to see that our changes and responses have resonated with you. We value your suggestion on the analysis of the performance of $\mu$Code-R and are looking at alternative approaches to combining the datasets. We will add this discussion in the next version of the paper in addition to your suggested changes for the theoretical analysis, wording/notation, and hyperparameters for reproducibility.
> > >
> > > All the best,\
> > > The Authors

---

### Official Review · Reviewer_QsKU · 2025-03-14

**Overall Recommendation:** 2

**Summary:**

The paper introduces an approach, $\mu\mathrm{CODE}$, for multi-turn code generation using single-step rewards. Unlike existing methods that rely on reinforcement learning, $\mu\mathrm{CODE}$ considers code generation as a one-step recoverable Markov Decision Process (MDP), allowing for iterative improvement through imitation learning. The method iteratively trains a generator and a verifier, where the verifier provides feedback to guide the generator in refining code solutions. Experimental results on MBPP and HumanEval datasets demonstrate that $\mu\mathrm{CODE}$ outperforms existing methods such as STaR.

## update after rebuttal
I appreciate the rebuttal by authors. However, my concerns were not well addressed.
1. While the paper claims that multi-turn RL methods suffer from sparse learning signals which makes learning inefficient, it fails to provide any comparative experimental evidence.
2. The authors claim that when using sparse rewards, the agent could not updated for responses where it did not generate any correct solution. But in this scenario, the verifier cannot receive effective training, therefore the continuous signals could also be quite biased. Using answers with the highest reward labeled by the verifier but actually incorrect as SFT labels could be potentially harmful.
3. I appreciate that the authors have provided more experimental results.

In summary, I would like to raise my score to weak reject.

**Claims And Evidence:**

No.

1. While the paper claims that RL methods are inefficient, it fails to provide comparative experimental evidence.

2. In the proposed iterative training process, the generator is trained on the verifier's annotated optimal results rather than the ground truth validated through actual code execution. This approach appears methodologically questionable, as verifier annotations may contain inherent biases. The rationale behind this design choice remains unclear.

3. The study's conclusions are drawn from experiments conducted exclusively on MBPP, a relatively small dataset of only several hundred samples. This limited scope, essentially amounting to a toy dataset, raises concerns about the robustness and generalizability of the findings. The conclusions, therefore, warrant further validation on more comprehensive datasets.

**Essential References Not Discussed:**

No.

**Experimental Designs Or Analyses:**

Yes.

**Methods And Evaluation Criteria:**

No.

**Other Comments Or Suggestions:**

No.

**Other Strengths And Weaknesses:**

Other Strengths: This paper is mostly well-written and clearly organized.

**Questions For Authors:**

Please refer to the content in Claims And Evidence.

**Relation To Broader Scientific Literature:**

Not related.

**Theoretical Claims:**

Yes.

---

> ### Author Rebuttal · Authors · 2025-04-01
>
> We thank the reviewer for their insightful feedback and are glad that you found the paper well organized. We are grateful for your questions and comments, and respond to your questions below.
>
> > While the paper claims that RL methods are inefficient, it fails to provide comparative experimental evidence.
>
> Most approaches for finetuning LLM agents via RL rely on rule-based verifier like execution feedback in code or correctness in math problems. Such signals are sparse and credit assignment due to long-term rewards is challenging in multi-turn tasks. We propose $\mu$Code that leverages a learned verifier to provide dense signals and also uses the one-step recoverability property for better credit assignment.
>
> *Sparse Rewards:*
> Let's consider a scenario where the agent is trained over the outcome of code execution. In such a setting, the reward can be either 0 or 1 denoting failure or success. When a correct solution is observed, an RL algorithm like PPO would increase the likelihood of this response. However, the agent is not updated for responses where it did not generate any correct solution for the prompt while sampling, and thereby the completions are discarded. This makes outcome based RL inefficient as the model updates relies on generating a correct solution, and exploration to a correct solution can be challenging. In many RL applications, this is mitigated with a dense reward signal, and $\mu$Code leverages a learned verifier for training. The generator is updated to predict solutions with the highest reward and with training should climb uphill this reward function. To demonstrate the benefits, we compare Multi-STaR^2 which is similar to $\mu$Code and performs relabeling with the sparse and ground truth reward (in Table 1 of the paper), and observed that $\mu$Code trained with dense rewards performed better.
>
> *Long-term Rewards*:
> During RL training, the agent is trained to optimize the long-term rewards of the trajectories in multi-turn tasks. With the insight of one-step recoverability, we propose to relabel trajectories and improve credit assignment. To show this, we propose a baseline, Rejection Fine-Tuning, where we finetune the agent with the top rollouts ranked via the learned verifier (and call it RFT (LV)). We did a hyperparameter search over different number of top rollouts for this baseline and found picking top 6 solutions from 15 rollouts worked best. We observe that RFT (LV) does not perform well when compared with $\mu$Code.
>
> ### MBPP
> | Algorithm   | Pass@3 | BoN@3      |
> | :---| :------: | :------: |
> | Llama-3.2-1B-Instruct        | 37.3    | 42.7
> | RFT (LV)    | 30.6    | 37.1
> | $\mu$Code   | **42.2**    | **47.2**
>
> ### HumanEval
> | Algorithm   | Pass@3 | BoN@3      |
> | :-----------| :------: | :------: |
> | Llama-3.2-1B-Instruct        | 31.5    | 32.9
> | RFT (LV)    | 27.0    | 38.6
> | $\mu$Code   | **40.0**   | **45.5**
>
> > Verifier annotations may contain inherent biases.
>
> We agree that the agent can exploit the inaccuracies within the learned verifier. This happens because the distribution of reponses from the generator shifts with training and the learned verifier can return inaccurate reward values. To account for the covariate shift, $\mu$Code updates the learned verifier after every iteration. Since the ground truth reward is already known (from the execution feedback), the verifier is updated using the oracle outcome. We would like to highlight that the learned verifier in $\mu$Code is trained with the concatenated dataset from prior iterations. In our experiments, we also observed that updating the verifier across iterations is beneficial for learning.
>
> We additionally present emperical results with the generator of $\mu$Code and the learned verifier at the last iteration vs the learned verifier at iteration 0 (on rollouts from Instruct model). We observe upto 2% drops in performance on BoN@3 when using the verifier trained on Instruct rollouts vs the verifier updated with rollouts from current generator.
>
> | Verifier    | MBPP     | HumanEval|
> | :-----------| :------: | :------: |
> | LV (Iter 0)   | 46.8    | 42.5
> | LV (Iter 2)   | **48.1**    | **44.5**
>
>
> > The conclusions, therefore, warrant further validation on more comprehensive datasets.
>
> We conduct an experiment with the CodeContests dataset. We curated a 1K-sized dataset with randomly selected problems from the training set of CodeContests. We use Llama-3.1-8B-Instruct and compare with MultiSTaR and $\mu$Code which were trained for 1 iteration on 4 X A100 GPUs with 80GB memory. We report the Pass@3 and BoN@3 results on the test set in CodeContests with 165 problems. We observe that $\mu$Code outperforms Multi-STaR by 1.6% on BoN scores showing the applicability of $\mu$Code to more challenging benchmarks.
>
> | Method  | Pass@3 | BoN@3 |
> | :---- | :----: | :----: |
> | Llama-3.1-8B-Instruct        |   4.4      | 7.4
> | Multi-STaR  |   6.8      | 10.3
> | $\mu$Code   |   **8.2**      | **11.9**
>
> Best Regards,
>
> The Authors

---

### Official Review · Reviewer_hcVQ · 2025-03-14

**Overall Recommendation:** 3

**Summary:**

The paper introduces $\mu$-Code, a scalable approach for multi-turn code generation that utilizes single-step rewards. It treats code generation as a one-step recoverable MDP, allowing correct code recovery from any intermediate state, and integrates a generator and verifier to iteratively train the system. The approach demonstrates substantial improvements over existing methods like STaR, with a detailed analysis of reward models and policy design choices.

**Claims And Evidence:**

I believe that most of the claims are well supported by experiments and theory. One debatable point is whether multi-turn code generation can be considered a one-step recoverable MDP. In a reasonable trajectory, each step is usually a gradual improvement over the previous one, for example, correcting the error in the last step. If a step is significantly wrong, it is difficult for the next step to directly recover to the correct code. I would like to see a more in-depth discussion regarding this claim, rather than treating it merely as a basic assumption.

**Essential References Not Discussed:**

I did not identify essential references not discussed.

**Experimental Designs Or Analyses:**

The experimental results are puzzling. HumanEval consists of 164 problems, and when measuring the pass rate, each problem is either correct or incorrect, meaning the metric must be a multiple of 1/164 ≈ 0.6%. However, most of the reported results in Table 1 for HumanEval do not yield integers when multiplied by 164. Given that the paper does not provide the code, this raises concerns about the validity of the reported results.

**Methods And Evaluation Criteria:**

- For the methodology section, I did not identify any issue and believe it makes sense for the problem and aligns with the one-step recoverable MDP assumption.

- For the experiments, HumanEval and MBPP are somewhat outdated at present. I recommend that the authors use Evalplus' HumanEval+ and MBPP+. Additionally, utilizing more practical code generation benchmarks such as BigCodeBench could further enhance the robustness of the paper.

**Other Comments Or Suggestions:**

The paper contains many typos. I suggest the authors thoroughly proofread the paper again.

- Algorithm 1, line 2: {(x, st, yt, ot))} -> {(x, st, yt, ot)}; line 4: trained on -> on

- Line 200 (right): during training?, -> during training?

**Other Strengths And Weaknesses:**

Strengths:

- The proposed method is simple and reasonable, fitting well with the one-step MDP assumption.
- The authors conducted solid theoretical analysis of the proposed method, deriving a performance bound.
- The writing is good and easy to follow.

Weaknesses:

The paper does not appear to provide source code or explicitly commit to releasing it (unless the reviewer has overlooked this information). The absence of publicly available code could impact the reproducibility of the results and weaken the work's value to the broader research community.

**Questions For Authors:**

- In Appendix A.2, the authors manually divided the test cases for HumanEval and MBPP. Could you consider using the additional tests introduced by HumanEval+ and MBPP+ as a private test set?

- Can the method described in this paper be extended to scenarios beyond code generation to construct trajectories?

- During the inference phase, if the verifier is not used and instead only the public test is used to select the best solution, will the performance change?

**Relation To Broader Scientific Literature:**

The method described in this paper may be extended to scenarios beyond code generation to construct trajectories.

**Theoretical Claims:**

I did not check the proof for the theoretical claims since I am not an expert on RL. The explanation of the theorems makes sense for me.

---

> ### Author Rebuttal · Authors · 2025-04-01
>
> We thank the reviewer for the valuable feedback and are glad that you found our approach simple and theoretical justification insightful. We address your questions below.
>
> > I would like to see a more in-depth discussion regarding this claim, rather than treating it merely as a basic assumption.
>
> For the task of code generation, there are two major sources of errors: 1) syntax errors and 2) algorithmic errors. For (1), we expect gradual improvement because fixing the errors should lead to a better solution at the next turn. However, the change might not be gradual for (2) when the approach is incorrect. In this scenario the agent might have to modify the algorithm and as pointed out by the reviewer can be difficult to recover from. Since, the task of code generation is one-step recoverable, this allows $\mu$Code to relabel with a better solution and thereby recover from harder scenarios. In the paper, we provided the theoretical justification to support this claim in Theorem 3.2 (Section 3.5) and our experiments demonstrate the efficacy of $\mu$Code. Below, we present an example where $\mu$Code successfully resolves an algorithmic error.
>
> **Prompt**: Write a function to calculate the height of the given binary tree.
>
> **Turn 1**
> ```
> def max_height(root):
>     if root is None:
>         return 0
>
>     queue = [(root, 1)]
>
>     while queue:
>         node, height = queue.pop(0)
>         if node is None:
>             return height
>         queue.append((node.left, height + 1))
>         queue.append((node.right, height + 1))
>
>     return -1
> ```
>
> **Turn 2**
> ```
> def max_height(root):
>     if root is None:
>         return 0
>
>     left_height = max_height(root.left)
>     right_height = max_height(root.right)
>
>     if left_height > right_height:
>         return left_height + 1
>     else:
>         return right_height + 1
> ```
> Note that the first solution takes an iterative approach and is incorrect because the function returns as soon as any leaf node is reached. However, the code solution at next iteration switches from an iterative to a simpler recursive approach and is correct.
>
> > Reproducibility
>
> We will be releasing the codebase, model checkpoints and hyperparameters with the final version of the paper.
>
> > Can the method described in this paper be extended to scenarios beyond code generation to construct trajectories?
>
> Yes, our method can be extended to any tasks where the correct solution can be generated in a single step (one-step recoverability). An application is theorem proving, where the the agent needs to write proof of theorems and the solution can be verified using external tools [1,2].
>
> > Could you consider using the additional tests introduced by HumanEval+ and MBPP+ as a private test set?
>
> We thank the reviewer for bringing this up. In the submitted version, we used the HumanEval and MBPP datasets with its own set of public and private test split. Below, we provide comparison where private tests include the additional tests provided in MBPP+ and HumanEval+. We report results with the same model 1B-sized model checkpoints used to report results in Table 1 and present the results on Pass@3 and BoN@3.
>
> ### MBPP+
> | Algorithm              | Pass@3 | BoN@3 |
> | :---------------- | :------: | :------: |
> | Llama-3.2-1B-Instruct        |43.2| 49.9
> | Multi-STaR  |44.3| 50.0
> | $\mu$Code   |**48.7**| **55.1**
>
> ### HumanEval+
> | Algorithm              | Pass@3 | BoN@3 |
> | :---------------- | :------: | :------: |
> | Llama-3.2-1B-Instruct        |25.7| 31.9
> | Multi-STaR  |26.7| 34.3
> | $\mu$Code   |**32.4**| **40.0**
>
> We observe that $\mu$Code outperforms baselines Multi-STaR by 5.1% and 5.7% on BoN@3 on MBPP+ and HumanEval+ datasets. Note that the numbers for MBPP are higher than what is reported in Table 1 because MBPP+ comprises of 378 problems as compared to 500 problems in the MBPP dataset.
>
> > Bigger benchmark
>
> We conduct an experiment on CodeContests dataset and show that $\mu$Code outperforms Llama-3.1-8B-Instruct and MultiSTaR. The experiment is discussed in the response to Reviewer QsKU, where we show **1.6%** performance gain over Multi-STaR at BoN@3.
>
> > Most of the reported results in Table 1 for HumanEval do not yield integers when multiplied by 164.
>
> We are using a temperature of 0.7 for sampling to report our metrics. Because of the smaller dataset size, we observed variance in the results and resorted to generating 15 rollouts for each prompt. We report the mean scores in Table 1 and the values in Table 1 will be a multiple of .6 / 15 = 0.04. In the revised version, we plan to add the standard error with the mean results for clarity.
>
> Best Regards,
>
> The Authors
>
> ### References
> [1] STP: Self-play LLM Theorem Provers with Iterative Conjecturing and Proving, Dong el al. 2025.
>
> [2] Learning Formal Mathematics From Intrinsic Motivation, Poesia et al., NeurIPS 2024.

---

> > ### Comment · Reviewer_hcVQ · 2025-04-02
> >
> > Thanks for your detailed response. The rebuttal has addressed most of my concerns. I decide to keep my positive score.

---

> > > ### Author Response · Authors · 2025-04-08
> > >
> > > Reviewer hcVQ,
> > >
> > > We would like to thank you for your support which helped improve the paper. We are thrilled to see that our answers have resonated with you and are happy to any further questions you may have.
> > >
> > > Best Regards,\
> > > The Authors

---

### Official Review · Reviewer_vyX7 · 2025-03-15

**Overall Recommendation:** 4

**Summary:**

This paper addresses code generation from multi-turn execution feedback. The authors propose uCode a simple and scalable approach that solves multi-turn code generation using only single-step rewards. The key insight is that code generation is a one-step recoverable MDP. uCode trains both a generator and a verifier. Experiments show improvement against StAR.


## update after rebuttal: Thank you for your answer. I have increased my score.

**Claims And Evidence:**

1. Novel framework
- Yes
2. multi-turn best-of-n approach that utilizes a learned verifier at inference time
- Yes (although it does not seem necessarily novel)
3. theoretical analysis
- Yes
4. Experiments on MBPP and Human Eval
- Yes
5. Analysis
- Yes

**Essential References Not Discussed:**

No

**Experimental Designs Or Analyses:**

The models are Llama-3.2-1B instruct or 8B instruct. I would recommend the authors to also investigate other LLMs that are not Llama, and include larger models. Turn limit = 3. Datasets and metrics used are the standard ones. Fundamentally, the only baseline used is STaR. I would encourage the authors to include another model.

Other than that, the experiments are well executed and the analysis are sound!

**Methods And Evaluation Criteria:**

The main idea is that one-step recoverable MDP implies that a correct code can be recovered from any intermediate state in a single step. uCode follows an expert iteration framework with a local search expert. It trains iteratively two components: a learned verified and generator. At inference time, both are used as best-of-n search.

First, uCore needs a dataset that is created using the current generator. The verifier is trained on the aggregated dataset. An expert is constructed and the dataset is re-labeled. Finally, the model is fine-tuned using the dataset. Repeat as many time until convergence. The training part is sound. At inference, the authors propose multi-turn best-of-N. At each turn, the generator produces N one-step rollouts and the verifier selects the most promising one and is executed in the environment to obtain the feedback. The process is repeated.

Finally, the theoretical analysis is sound.

**Other Comments Or Suggestions:**

No

**Other Strengths And Weaknesses:**

Overall, well-written paper and well structure. Easy to read and follow. The methodology is sound, and the experiments as well. I would encourage the authors to include another baseline investigate the performance when the turn limit is above 3.

**Questions For Authors:**

What are the performance with another LLM that is not Llama? Same with bigger model.
Could you provide a performance analysis with different turn limits (> 3)?
Same with the "training" time following algorithm 1, and cost/efficiency.

**Relation To Broader Scientific Literature:**

I am not familiar with the literature.

**Theoretical Claims:**

Yes the one in section 3.5

---

> ### Author Rebuttal · Authors · 2025-04-01
>
> Thank you for your constructive feedback. We are glad that you found the paper well structured and our experiments thorough. We address your questions below.
>
> > Investigate other LLMs that are not Llama, and include larger models.
>
> We investigate the performance of the Qwen-2.5-1.5-Instruct model, Multi-STaR and $\mu$Code at Pass@3 and BoN@3 metrics on MBPP and HumanEval datasets. Below, we present the table and show that $\mu$Code outperforms the baselines.
>
> ### MBPP
> | Algorithm         | Pass@3 | BoN@3 |
> | :---------------- | :------: | :------: |
> | Qwen-2.5-1.5-Instruct    |53.8|60.9
> | Multi-STaR  |57.0|58.0
> | $\mu$Code   |**59.0**|**63.1**
>
> ### HumanEval
> | Algorithm         | Pass@3 | BoN@3 |
> | :---------------- | :------: | :------: |
> | Qwen-2.5-1.5-Instruct    |64.5|70.3
> | Multi-STaR  |66.6|71.3
> | $\mu$Code   |**70.5**|**74.0**
>
> Our experiments are restricted to models that can be trained on 4 x A100 GPUs with 80 GB memory each and training of larger models greater than 8B requires more GPU memory. We expect $\mu$Code should scale to larger models with more compute.
>
> > Investigate the performance when the turn limit is above 3. > Same with the "training" time following algorithm 1.
>
> In the paper, we ran BoN search for a turn limit of 3. We present the results with a turn limit of 6 below (with the 1B-sized model used in Table 1). Note that the model is trained with a turn limit of 3. $\mu$Code achieves the best performance on both datasets across most turn limits. This highlights the versatility of $\mu$Code at generalizing to larger turn limits at test-time. Additionally, our results show that $\mu$Code achieves the largest gains per turn, suggesting it is better at incorporating execution feedback and improving code solutions across turns. This experiment validates that $\mu$Code is performant with more turns during evaluation. We believe that training with a higher turn limit could further improve the performance.
>
> ### MBPP
> | Algorithm         | T=1 | T=2 | T=3 | T=4 | T=5 | T=6
> | :---------------- | :------: | :------: | :------: | :------: | :------: | :------: |
> | Llama-3.2-1B-Instruct        |39.2|41.5|43.1|44.0|45.0|45.5|
> | Multi-STaR  |**39.4**|40.9|41.4|41.9|42.2|42.7|
> | $\mu$Code   |38.1|**43.9**|**46.6**|**48.8**|**50.1**|**50.9**|
>
> ### HumanEval
> | Algorithm         | T=1 | T=2 | T=3 | T=4 | T=5 | T=6
> | :---------------- | :------: | :------: | :------: | :------: | :------: | :------: |
> | Llama-3.2-1B-Instruct        |34.8|35.6|36.6|38.0|38.4|38.6|
> | Multi-STaR  |36.8|36.8|37.2|37.8|38.2|38.2|
> | $\mu$Code   |**38.2**|**45.7**|**48.4**|**50.6**|**51.0**|**51.4**|
>
> > Fundamentally, the only baseline used is STaR.
>
> We point the reviewer to our response to Reviewer QsKU where we have provided an additional baseline: Rejection Fine-tuning with a learned verifier, or RFT (LV). Here, we select the top trajectories ranked using the learned verifier for a prompt and finetune at each iteration. This is similar to the Multi-STaR baseline, where the learned verifier is used to filter trajectories instead of the ground truth reward. We observed that $\mu$Code performs better than RFT (LV) baseline.
>
> Best regards,\
> The Authors

---

> > ### Comment · Reviewer_vyX7 · 2025-04-03
> >
> > Thank you for your answers. I will keep my score as is.

---

> > > ### Author Response · Authors · 2025-04-08
> > >
> > > Dear Reviewer vyX7,
> > >
> > > We sincerely thank you for your feedback and are delighted to see that our experiments and responses have resonated with you.
> > >
> > > All the best,\
> > > The Authors

---

### Decision · Program_Chairs · 2025-05-01

**Decision:**

Accept (spotlight poster)

**Comment:**

In this paper, the authors propose a new approach to improve multi-turn code generation using single-step rewards. Specifically, they formulate the code generation as a one-step recoverable MDP, where the correct code solution can be generated from any intermediate state in a multi-turn trajectory. A generator is trained to generate code solutions using multi-turn execution feedback and a verify is used to score the generated code.

There are some minor concerns raised by the reviewers:
- The authors limited the experiments to Llama models with max turns up to 3 on HumanEval and MBPP. The authors added additional results in the rebuttal to extend the experimental setup (larger number of turns, results on other benchmarks)
- Missing details of experimental configuration in the main paper. For reproducibility, the authors responded with additional information on the training setup and they will release the codebase and model checkpoints.

Overall, the proposed method is well motivated with sound techniques, demonstrated by comprehensive experimental results. I suggest the authors review all the feedback and improve their paper accordingly.